# History, Phylogeny, Biodiversity, and New Computer-Based Tools for Efficient Micropropagation and Conservation of Pistachio (*Pistacia* spp.) Germplasm

**DOI:** 10.3390/plants12020323

**Published:** 2023-01-10

**Authors:** Esmaeil Nezami, Pedro P. Gallego

**Affiliations:** 1Department of Plant Breeding, Agriculture Research School, Nuclear Science and Technology Research Institute (NSTRI), Karaj P.O. Box 31485-498, Iran; 2Department of Plant Biology and Soil Science, Faculty of Biology, University of Vigo, 36310 Vigo, Spain

**Keywords:** artificial neural networks, cultivars, cryopreservation, cuttings, design of experiments, domestication, etymology, grafting, machine learning, molecular characterization, *Pistacia vera*, plant tissue culture, rootstocks, taxonomy

## Abstract

The word “pstk” [pistag], used in the ancient Persian language, is the linguistic root from which the current name “pistachio”, used worldwide, derives. The word pistachio is generally used to designate the plants and fruits of a single species: *Pistacia vera* L. Both the plant and its fruits have been used by mankind for thousands of years, specifically the consumption of its fruits by Neanderthals has been dated to about 300,000 years ago. Native to southern Central Asia (including northern Afghanistan and northeastern Iran), its domestication and cultivation occurred about 3000 years ago in this region, spreading to the rest of the Mediterranean basin during the Middle Ages and finally being exported to America and Australia at the end of the 19th century. The edible pistachio is an excellent source of unsaturated fatty acids, carbohydrates, proteins, dietary fiber, vitamins, minerals and bioactive phenolic compounds that help promote human health through their antioxidant capacity and biological activities. The distribution and genetic diversity of wild and domesticated pistachios have been declining due to increasing population pressure and climatic changes, which have destroyed natural pistachio habitats, and the monoculture of selected cultivars. As a result, the current world pistachio industry relies mainly on a very small number of commercial cultivars and rootstocks. In this review we discuss and summarize the current status of: etymology, origin, domestication, taxonomy and phylogeny by molecular analysis (RAPID, RFLP, AFLP, SSR, ISSR, IRAP, eSSR), main characteristics and world production, germplasm biodiversity, main cultivars and rootstocks, current conservation strategies of both conventional propagation (seeds, cutting, and grafting), and non-conventional propagation methods (cryopreservation, slow growth storage, synthetic seed techniques and micropropagation) and the application of computational tools (Design of Experiments (DoE) and Machine Learning: Artificial Neural Networks, Fuzzy logic and Genetic Algorithms) to design efficient micropropagation protocols for the genus *Pistacia*.

## 1. Introduction

This review is a comprehensive summary of the most recent scientific literature on the state of pistachio history, biodiversity, production and how unconventional pistachio propagation systems, combined with artificial intelligence tools, can help conserving *Pistacia* spp. germplasm while preventing its genetic erosion and, at the same time, satisfy many curiosities unknown to most pistachio researchers, growers, consumers and those interested in pistachio.

The etymology of the pistachio’s name originates from only one word ‘pstk’ [pistag] in the ancient Persian language, Avestan [1]. ‘Pista’ constitutes the root for the word pistachio in almost all languages that arose from the Persian and from the oldest Central Asia classical languages such as ‘pista’, ‘psta’, ‘pisse’ or ‘miste’ and the Mediterranean countries such as ‘pistákia’, ‘pistacia’, ‘fistuk’, ‘fustak’ or ‘fistik’ and in all present-day modern languages derived from those classical languages such as ‘pistacchio’, ‘ pistachio’, ‘pistacho’, ‘pistache’ or ‘pistazie’ [2,3]. At present, although the word pistachio is applied to several species (plants and fruits) of the genus *Pistacia*, only the edible fruits of the species *Pistacia vera* L., which is most widely accepted by consumers, are marketed under the name pistachio worldwide.

The origin of the pistachio plant was not clear for a long time. Initially, it was thought that pistachio was native to the arid areas of old Mesopotamia, currently Syria and Iraq, and some parts of Turkey and Iran; however, later studies have suggested that pistachio was first consumed by Neandertals around 300,000 years ago, and later (around 8000 years BC) in southern Central Asia [4], including northern Afghanistan and northeastern Iran. The current evidence suggests that the domestication and cultivation of the plant occurred around 3000 years ago in Iran, introduced firstly to Syria and later to Italy during the Roman period, expanding to the rest of the Mediterranean basin during the Middle Ages [5] and finally imported to the USA, the largest pistachio producer country, in the late 1880s [3,6].

Pistachio is by far the most productive and economically important species of edible nuts, well-known for its large, delicious seeds which are eaten whole (salted, roasted, or fresh). Each adult tree can produce around 50 kg of seeds per year.

The Persians called the pistachio fruit ‘the smiling nut’ and the Chinese name for pistachio ‘hoi sum guo’, means “happy fruit”; in both cases because the shelled fruit closely resembles a human smile; a fact that invites us to consume the pistachio, but better to do it with the shell than without it. In fact, some researchers [7] found that leaving the shell on the pistachio as a visual cue significantly (*p*-value < 0.05) helped to ingest fewer pistachios, and therefore fewer calories, than if pistachios were consumed without the shell and therefore the amount of empty shells was not visible. In any case, every 100 g of seed is a good source of calories (562 kcal).

Pistachio nuts are composed mainly of lipids (46 g) mostly unsaturated fatty acids (29 g), carbohydrates (28 g), proteins (21 g), and dietary fiber (10 g), but are also a rich source of twelve vitamins and several minerals, being rich in potassium (over 1 g), and phosphorus (0.5 g) and in bioactive phenolic compounds such as total phenolics, carotenoids (lutein and zeaxanthin), phytosterols and, as dry nuts, 4% water [6,8]. These components of the pistachio kernel help to promote human health due to their antioxidant capacity and oxidative stability, anti-inflammatory properties, anti-Crohn’s diseases, antifungal, antimicrobial, and antiviral, anti-anxiety and depressive-like behaviors, and finally, modulation of the gut microbiota [9,10,11].

Pistachio is considered one of the main economical crops of countries located in semi-desert and desert regions of those countries placed in the Middle East, Central Asia, Mediterranean Sea, and the USA (mainly in California). Approximately one million t of pistachios are harvested worldwide, and of them almost 98.64% were produced in just five countries: USA, Turkey, Iran, China and Syria [12]. Another five countries were the largest importer: Germany, Italy, India, Spain, and Belgium have 59.70% of the total importations [12]. USA and Iran have 82% of the world pistachio exports and are also the top countries by consumption. China, USA, Turkey, Iran, Germany, Syria, Spain, Vietnam, Italy, India, France, and Afghanistan representing 71.40% of total world consumption [13].

Most of the current pistachio species are not normally cultivated and grow in the wild. They are used for local fruit consumption, oil extraction, and soap production [14]. The largest proportion of pistachio production is limited to very few and specific high yielding commercial cultivars such as the female ‘Kerman’ of Iranian origin [15].

Distribution and genetic diversity of wild and domesticated pistachio plants have been diminishing due to increasing human demographic pressure and to climate change which have destroyed the natural habitats of pistachio trees, and the monoculture of selected cultivars. However, most of the genetic resources of *Pistacia* sp. may have become extinct, as is the case for relict varieties such as “Natalora”, “Rappa di sessa”, “Hinnulina” and “Agostina”, as they are no longer located in present-day pistachio orchards [16]. There are different causes that have led to a continuous and extreme genetic erosion observed in pistachio trees: (1) the rapid spread of a few high-yielding commercial pistachio varieties [17]; the destruction of natural pistachio habitat [18] due to anthropogenic pressure, such as unregulated forest clearing for agricultural purposes, causing intentional fires for those or other industrial purposes [19] and; (2) due to the narrow genetic base of all those selected cultivars that limit their tolerance to biotic (emergence of new diseases and pests) and abiotic (climate change) stresses [2].

*Pistacia* spp. germplasms were traditionally conserved and multiplicated via conventional macropropagation techniques such as seed germination, cuttings and grafting using in situ and ex situ strategies. More recently, unconventional biological techniques (cryopreservation, slow-growth storage, synthetic seeds and micropropagation) have opened new strategies for preservation of commercial and endangered *Pistacia* species [19,20,21,22,23]. However, many of those low efficient conventional and unconventional procedures can be changed for new highly efficient protocols, based on the combination of biotechnological and computer-based tools for successful pistachio propagation.

In summary, the current status of the etymology, origin, domestication, taxonomy and phylogeny, rootstocks, germplasm biodiversity and the most advanced plant biotechnological and computer-based tools (DoE and Machine Learning) to design efficient plant tissue propagation methods for efficient conservation of pistachio germplasm, have been discussed and summarized in this comprehensive review.

## 2. Pistachio History

### Pistacia Etymology, Origin and Domestication

Pistachio has many names but only one root. ‘Pista’ constitutes the root for the word pistachio in many European languages; exported from the Persian Empire to the Classical period in Greek ‘pistákia’ [pistakeon] and latter in Latin ‘pistacia’ from which it derives the current names of ‘pistacchio’ (Italian), ‘pistachio’ (English), ‘pistacho’ (Spanish), ‘pistache’ (French), ‘pistazie’ (German); but also in many other Central Asia such as ‘pista’ in Uzbek and Tajik, ‘psta’ in Kazakh, ‘pisse’ in Turkmen, ‘miste’ in Kyrgyz; and even ‘fistashka’ (Russian), ‘fistuk’(Hebrew), ‘fustak’ (Arabic), ‘fistik’ (Turkish) which comes the from ‘pstk’ word [2,3]. Nowadays, the pistachio word is applied to the plant and fruits of *P. vera* L.

*Pistacia* is believed to have originated in central Asia at two centers of diversity: one was the Mediterranean region of Southern Europe, Northern Africa, and West Asia, while the second species of the genus occurs naturally from North Africa to East Asia (Philippines) and America (from Texas to Nicaragua) [24].

The origin of the pistachio was not clear for a long time. Initially, it was thought that pistachio was native to the arid areas the old Mesopotamia, currently Syria and Iraq, and some parts of Turkey and Iran [25]. However, in the early part of the 1990s, an expedition of Russian botanists described an area of wild *P. vera* in southern Central Asia: Turkmenistan, Uzbekistan, Tajikistan, Kazakhstan, and Kyrgyzstan [2], including northern Afghanistan and northeastern Iran [26].

Currently, *Pistacia* sp. are distributed among five more-or-less isolated floristic regions including countries located in semi-desert and desert habitats: (i) Mediterranean region; (ii) Irano-Turanian region; (iii) North East Highlands region; (iv) Sino—Japan region; and (v) Mexican regions [27].

Furthermore, fossil evidence suggested that *Pistacia* sp. evolved around 80 million years ago [26]. More recent genetic studies have demonstrated a common ancestor for all pistachio species at between 1.5 and 5.0 million years old, suggesting that pistachio evolved much more slowly than other species and that the most ancestral cultured species, *P. vera* L. is 1.1–3.7 million old, being the most evolved species *P. terebinthus* L. [28].

Archaeological studies have found that the pistachio was first consumed by Neandertals around 300,000 years ago [29,30] and was as common food source for humans around 8000 years ago [4], not only in southern Central Asia but also in Mesopotamia, as revealed by the presence of pistachio nuts in a dig site at Jarmo, near northeastern Iraq [31]. More recent discoveries have dated the first accurate pistachio consumption in Djarkutan (Uzbekistan) in 3500–1700 B.C., during the Bronze Age [32].

Current scientific evidence proposes that the domestication and cultivation of pistachio occurred around 3000–4000 years ago in the Persian Empire [6], extended westward to the Mediterranean countries [26], firstly to Syria (1000 B.C. and 1000 A.D.) and later to Italy during the Roman period. Dioscorides [11], in the 1st century A.D., in the book ‘De Materia Medica, 1,124,1,1’, described pistachios as being produced in Syria and stated that they had medical properties. Around 30 A.D., according to Chitzanidis [25]: ‘Pliny the elder (Natural History, XV, 83, 91) the knight Vitelius brought the pistachio to his house in Rome at the end of the reign of Tiberius and in the same year the knight Flacus Pompeius introduced it into Spain’ [33]. During the Middle Ages, pistachio cultivation expanded to the rest of the Mediterranean basin [5], and in the 19th century, pistachio was introduced in the USA and Australia [3,14,34].

However, historical records have revealed additional information about the presence of pistachio trees in some places where none exist today due to genetic erosion by the neglectful underutilization of these valuable genetic resources [35], and water deficiencies caused by climate change [36]. The present distribution has been impacted by the exploitation of pistachio trees by local human populations, who have used them as a source of fuel and heavy pasturing of cattle, preventing natural renewal [37], and replacing them with a few high-yield commercial cultivars. More recent details about the pistachio origin and cultivation can be found in recent reports [3,38].

## 3. Pistachio Taxonomy and Phylogeny

### 3.1. Pistacia Taxonomy

The *Pistacia* genus belongs to the Anacardiaceae (R.Br.) Lindl. family (order Sapindales) which initially included approximately 82 genera, comprising around 700 species of trees, shrubs, and lianas that produce a milky or watery sap, exude gums and resins and are generally native to pantropical (tropical and subtropical) countries [36], although some species are native to temperate regions. More recently, the number of species and genera have increased to 83 genera, comprising around 860 species of trees [39,40].

The next specifications distinguish the Anacardiaceae family from others of Sapindales [41,42]:An uncommonly high proportion of dioecy (86% of the genera).Small trees and shrubs (5–15 m), and woody vinesLeaves are usually compound, deciduous or evergreen, alternate and composed of leaflets in several arrangements.The small flowers are formed in branched clusters (panicles) and may be bisexual or unisexual. Main feature of the flower is a prominent nectar disk and a pistil of fused carpels forming a seed cavity in which an ovule develops.Fruits are usually drupes.

The genus *Pistacia* was described for the first time by Carl Linnaeus [43] who recognized six species: *P. lentiscus* L., *P. narbonensis* L., *P. simaruba* L., *P. terebinthus* L., *P. trifolia* L. and *P. vera* L. The genus *Pistacia* was taxonomically firstly included in the family Pistaciaceae by Adans in 1763 based on some morphological differences (flower structure, pollen and style) [44] but recent studies in both morphological and molecular aspects, have shown that it belongs to the family of Anacardiaceae [39], distinguished from other members of this family by its reduced flower structure, feathery (plumose) styles, and unusual pollen morphology [39].

In the first monograph study of *Pistacia* genus, Engler [45] listed eight species and a few subspecies; however, the taxonomical classification of *Pistacia* was started in the middle of the 19th century, when Zohary [26] subdivided *Pistacia* into four sections and 11 species based on leaf characteristics and nut morphology:(i)Lentiscella Zoh. (*P. mexicana* Humn., Bonpl. and Kunth and *P. texana* Swingle).(ii)Eu-*Lentiscus* Zoh. (*P. lentiscus* L., *P. saportae* Burnat. and *P. weinmannifolia* Poisson)(iii)Butmela Zoh. (*P. atlántica* Desf.)(iv)Eu-Terebinthus Zoh. (*P. chinensis* Bge. *P. khlinjuk* Stocks, *P. palestina* Boiss., *P terebinthus* L., and *P. vera* L.).

### 3.2. Pistacia Phylogeny

Those leaf characteristics and nut morphology have been regarded as one of the earliest, cost-effective and efficient way to identify species in the genus [46]. Even currently, a morphological dichotomous key is still widely used for rapid identification of different species of *Pistacia* on the basis of leaf characteristics [24]; particularly useful for pistachio growers who are not able to use complex DNA analysis (characteristics have been summarized in Table 1).

Between 1960 and 1990 [47,48,49,50,51], primary molecular markers on the basis of isozyme banding patterns opened new robust insights for characterizing the genetic variation present in several plant species (Figure 1). Using this methodology, Loukas and co-workers [52] successfully employed pollen isozyme polymorphism in identifying relationships between *P. vera*, *P. terebinthus* and *P. lenthiscus*, proposing a close relationship between *P. terebinthus* and *P. vera* rather than *P. lenthiscus* and *P. vera*. Although, several efforts were made towards classification of different species of *Pistacia* sp. [53] or within *P. vera* cultivars [17,54] using this methodology, the requirement for relatively large quantities of plant material to extract sufficient enzymes, time consuming issues and the inadequate coverage of genome data due to the low number of available isozymes markers, have been considered a main disadvantage of this scientific technique [55]. Those limitations encouraged researchers to develop a range of more informative and robust molecular markers with higher rates of polymorphism. Furthermore, molecular markers based on DNA are stable, detectable in all tissues and independent of environmental or seasonal conditions. They have been employed for multiple applications in plant science such as doubled haploid (DH) technology to the identification/assessment of ‘purity’/hybrid testing, genetic diversity analysis, genetic linkage map construction, mapping of quantitative trait loci (QTLs), map-based cloning of genes, mapping of mutations, marker-assisted selection (MAS), marker assisted backcross breeding (MAB), marker-assisted pyramiding, mapping major genes, and characterization of transformants [56,57,58,59].

In the case of the pistachio, precise efforts have been made in order to provide an insight of genetic variability of the genus *Pistacia,* mainly on the basis of such different techniques as Randomly Amplified Polymorphic DNA (RAPD) [60,61,62,63]; RFLPs [64], Amplified Fragment Length Polymorphisms (AFLPs) [63]; Inter Simple Sequence Repeat (ISSR) [65,66]; Simple Sequence Repeat (SSR) [67,68], Inter Retrotransposon Amplified Polymorphism (IRAP) [66,69], retrotransposon microsatellite amplified polymorphism (REAMP) [69], start codon targeted (SCoT) [70] and eSSR [71]. Figure 1 shows the related progression from emerging hybridization-based markers since 1980.

The earliest studies combining morphological characters and chloroplast genome (plastid restriction site) analysis [28] suggested the division of the *Pistacia* genus into just two sections: *Lentiscus* (including sects. *Lentiscella* and *Eu Lentiscus*) characterized by evergreen shrubs with paripinnate leaves and small seeds, and *Terebinthus* (including *Butmela* and *Eu Terebinthus*) characterized by deciduous trees with imparipinnate leaves and large seeds. Parfitt and Badenes [64] applied RFLP analysis to study the phylogenetic relationship of ten *Pistacia* cpDNA, using tobacco chloroplast DNA probes, pinpointing *P*. *vera* as the most ancient of the 10 species studied. Whilst *P*. *integerrima* and *P*. *chinensis* were shown to be distinct species, the pairs of species ‘*P*. *vera*: *P*. *khinjuk*’ and ‘*P*. *mexicana*: *P*. *texana*’ could not be separated by their method. Later, RAPD and AFLP techniques [27,60,62,64,65,72,73,74] were employed for categorization of *Pistacia* sp. accessions belonging to six species from geographically different locations in Mediterranean areas, clustering genus *Pistacia* sp. into two groups; (i) all the *P. lentiscus* accessions and (ii) containing all other accessions. Furthermore, the latter group was divided into two subgroups, one including *P. palaestina* and *P. terebinthus*; the other consisting of *P. atlantica*, *P. khinjuk* and *P. vera* [63]. Although a closely related species of *P. khinjuk*, *P. vera*, and *P. atlantica* has been commonly reported for a long time using different molecular approaches including AFLP and RAPD [27,72,73], recent finding have shown a contrast. For instance, the application of SCoT markers have indicated *P. vera* and *P. khinjuk* are the most closely related species, while *P. atlantica* and *P. vera* have the highest genetic differentiation [70].

In numerous scientific reports *Pistacia* sp. has also been classified into two sections; ‘New World’ species (*P. mexicana* and *P. texana*) vs. ‘Old World’ [27,28,63,75,76,77,78], species, whilst the results of recent studies have not supported this kind of classification pattern. Xie and co-workers [79] using both nuclear and chloroplast genes have indicated that besides being a clade sister between New and Old World species providing a clear resolution of the diversity, accessions of the eastern Asia species (*P. weinmannifolia* and *P. cuchuongenesis*) form a strongly supported clade sister with a clade of remaining Old World species, which is in contrast with the previous aforementioned classifications. The results from these authors may also serve as proof that the *Pistacia* species track their environmental niche on the basis of climate changes during the genus evolution, followed by the decisive role of current environmental conditions on the recent distributional pattern of *Pistacia* sp. [80,81]. Altogether, in spite of successful efforts carried out concerning the characterization of the genus *Pistacia* using morphological and molecular markers, a complete classification of this genus requires further studies [19].

SSR markers, because of their abundance and hypervariability, are widely regarded as an important class of DNA markers [82,83,84], and numerous SSR markers have been developed for the assessments of *Pistacia* species diversity [85]. For example, Topçu and co-workers [86] reported the transferability of 100 out of 110 SSR primer pairs in ten wild *Pistacia* species, pinpointing *Pistacia eurycarpa* with the highest amount of transferability, while *Pistacia texana* and *Pistacia lentiscus* with the lowest. These authors suggested the application of SSR markers for germplasm characterization, construction of genetic linkage maps and comparative mapping in the genus *Pistacia*. However, major limitations of SSR markers originate from their high variability with respect to other genomic regions, which in turn might not necessarily reflect genome-wide patterns of genetic diversity [87,88,89]. More recently, advancements in next-generation sequencing (NGS) techniques enabled researchers to use high-throughput genotyping platforms in *Pistacia* sp. [68,85]. As an examples, with the help of transcriptome sequencing of the *Pistacia* species, eSSR markers were employed for phylogenic studies [71] due to their high efficiency compared to conventional SSR. These authors reported that the closest species to *P. vera* was *P. khinjuk*, whilst *P. eurycarpa* was closer to *P. atlantica* than *P. khinjuk*. However, ‘*P. atlantica*:*P. mutica*’ and ‘*P. terebinthus*: *P. palaestina*’ pairs of species were not clearly separated from each other in their studies [71].

In conclusion, while the first morphological analysis of the *Pistacia* genus integrated just 11 species and 1 subspecies [26], more recent studies classified this genus into 12 taxa (nine species and five subspecies) integrated into two sections using molecular approaches (Table 1): (1) *Pistacia* (*P. atlantica* Desf., *P. chinensis* Bunge, *P. eurycarpa* Yalt., *P. khinjuk* Stocks, *P. terebinthus* L., and *P. vera* L.) and (2) *Lentiscella* (*P. lentiscus* L., *P. Mexicana* Humb., Bonpl. and Kunth, and *P. weinmannifolia* Poiss. ex Franch), and inter-specific hybrid *P. × saportae* of trees or shrubs. It is worth noting that some previous taxa classified as species are currently considered members of one of those 12 taxa: such as *P. malayana* Herderson and *P. cucphuongensis* Dai currently *P. weinmannifolia* Poiss. ex Franch; *P. aethiopica* Kokward renamed as *P. lenticus* subsp. *emargianta Engl.*); *P. palaestina* Boiss *renamed as P. terebinthus* subsp. *palaestina* Boiss) and *P. texana* Swingle (sister taxa of *P. mexicana*) [79]. Finally, the chloroplast DNA analysis has revealed that the interspecific *P. × saportae* Burnat has *P. lentiscus* as maternal and *P. terebinthus* as paternal parents [77].

## 4. Pistachio Characteristics and Production

### 4.1. Pistachio Characteristics

The pistachio word has been applied to the plant and fruits of *P. vera*, the only *Pistacia* species grown for commercial nut production worldwide. It is the only member of the *Pistacia* genus that gives edible and economically important nuts [90]. Other *Pistacia* sp. are xerophytic, diploid, deciduous and dioecious trees, where both male and female plants are required to produce fruit. The trees tolerate a wide range of temperature (e.g., from −20 °C to 45 °C) in long and warm summers, cold winters with low humidity ratio (RH < 35%) and no irrigation or rain during the summer period, although seedlings require a very long period to reach the budding stage [91]. Under natural conditions, *P. vera* does not develop a central tap root, but produces a highly branched root system with many fine roots that permit the tree to efficiently absorb water and nutrients from the soil. Leaves are pinnately-compound with broad, elliptical to round-ovate leaflets. Each leaf covers a single axillary bud with the ability of differentiation into inflorescences to produce female or male flowers. Male flowers are apetalous and have 4–5 anthers inserted on a disc. Female flowers are apetalous, subtended by 1–3 small bracts and 2–7 bracteoles and borne in racemes or panicles and have no nectarines. The species is wind-pollinated. The tree produces cluster fruits with nuts (classified as a semidry drupe) consisting of three parts: an exocarp, a fleshy mesocarp, and an endocarp that encompasses a seed. The pistachio endocarp or shell encloses a single oil-rich seed. The exocarp changes from green to white or white-purple color at maturity [92]. In contrast to wild pistachio drupes, which are usually roughly spherical and small in size (4–8 mm), the *P. vera* fruit are quite oblong and larger with average length of 14 mm (min. 10 and max. 20 mm), with an average width of 8.28 mm (min. 6.5 and −max. 11.6 mm) and an average diameter (7.4 mm) [3,36].

### 4.2. Pistachio Production

Pistachio is considered one of the main economical crops of Central and Eastern Asia, Mediterranean Sea (South Europa, Western Asia, and North Africa), and North America, according to latest Food and Agriculture Organization’s report [12]; a total of approximately 1,125,305 t of pistachios were harvested worldwide, with the USA (42%), Turkey (26%), Iran (17%), China (7%), and Syria (6%) producing 1,110,010 t, comprising around 98.64% of the world market. The largest importer countries are Germany (22%), Italy (9%), India (9%), Spain (6%), and Belgium (6%), accounting for 223,378 t, representing 59.70% of the total importations (FAOSTAT, 2020) Finally, the latest statistical yearbook about nut markets in the world, published by International Nut and Dried Fruit Council (INC) [13], reported that in 2021 USA and Iran had 82% of the world pistachio exports in 2019, with the top countries by consumption: China, USA, Turkey, Iran, Germany, Syria, Spain, Vietnam, Italy, India, France, and Afghanistan, accounting 480,375 t (71.40% of total).

## 5. Pistachio Biodiversity: Cultivars and Rootstocks

### 5.1. Pistachio Germplasm Biodiversity

There are still historical pistachio fields, hosting very old trees of pistachio (~700 up to 1800-year-old) in some countries such as Iran and Syria, which need to be safeguarded before the complete loss of these local varieties [37,93]. Distribution and genetic diversity of wild and domesticated pistachio plants have been diminishing for many years due to increasing human demographic pressure (using those trees as wood fuel) and to climate changes, provoking strong rains and dramatic increases in soil and air temperature, causing severe drought and fires, destroying the natural habitats on mountain slopes and causing landslides [94]. In addition, the emergence of new diseases, mainly Septoria leaf spot (*S. pistaciarum* and *S. pistacina*), Verticillium wilt (*V. dahliae*), Phytoptora crown (*Phytophtora* spp.) and Armillaria (*A. mellea*) and Fusarium (*Fusarium* spp.) root rot [95] and pests (such as *Megastigmus pistacia*) some fungal canker pathogens of pistachio (*Leptosillia pistacia* and *Cytospora pistacia*) [96], have been widely spread in the traditional orchards or natural plantings. All together pistachio monoculture has been boosted based on a few new cultivars and rootstocks which are tolerant to abiotic and biotic stress but provoking a continuous genetic erosion. This is the most serious threat to the traditional pistachio cultivation.

The International Plant Genetic Resources Institute (IPGRI), supported by the Consultative Group on International Agricultural Research (CGIAR) was established in 2001 to accelerate the conservation and use of genetic diversity of Underutilized Mediterranean Species (UMS) program. Of the main objectives of this program, characterization and conservation of *Pistacia* germplasm in seed banks and clonal collection are exemplified [16]. Biodiversity conservation of *Pistacia* species, therefore, are a global concern, leads to the establishment of national and international gene banks for the characterization and conservation of *Pistacia* genetic resources in different countries such as the ‘Pistachio Research Institute’ at Rafsanjan (Iran), ‘Pistachio Research Institute’ at Gaziantep (Turkey), Institute of Plant Breeding and Genetic Resources(IPB and GR) at Lamias (Greece), the Institute of Agrifood Research and Technology (IRTA) at Reus (Spain), California Pistachio Research Board (CPRB) at Fresno (USA). These organizations also contribute to the characterization and conservation of *Pistacia* germplasms [19,97]. The major concern about pistachio germplasm is due to the presence of different ecological conditions in Central and West Asia, and North Africa (CWANA) and the Mediterranean countries, which remain the main sources of genetic diversity in pistachio for the world. Thus, an international collaboration must be clearly defined in order to preserve the widest biodiversity of pistachio as possible. For instance, only in the Kerman province of Iran are more than 70 cultivars being growing [98].

### 5.2. Pistachio Cultivars

The largest proportion of pistachio production in those main producer countries is limited to very few and specific high yielding commercial cultivars such as ‘Kerman’, ‘Golden Hills’, ‘Lost Hills’, Kaleghouchi’ and ‘Aria’ (♀) and ‘Peters’ (♂) cultivars in the USA [99]; ‘Uzun’, ‘Kirmizi’, ‘Siirt’ (♀) and ‘Kaska’, ‘Ozturk’, ‘Uygur’, ‘Atli’ (♂) cultivars in Turkey [100,101]; as well as ‘Ashoury’, ‘Red Oleimy’ (♀) and ‘Arab-aleppo’ (♂) cultivars in Syria [102]. However, in Iran and in other countries a large part of the genetic resources of *Pistacia* sp. may have become extinct, as is the case for relict varieties such as ‘Natalora’, ‘Rappa di sessa’, ‘Hinnulina’ and ‘Agostina’, as they are no longer located in present-day pistachio orchards [16].

It is worth noting that most of the current pistachio species are not normally cultivated and grow in the wild; they are used for local fruit consumption and their oil is extracted for soap production [14]; the wood is made into charcoal for fuel for cooking ovens and for iron smelting; the resins, pitch and tannins are used for the production of a number of products such as lacquer, varnish and turpentine [2]. Despite this, there are still historical pistachio fields hosting very old trees of pistachio (~700 up to 1800-years-old) in some countries such as Iran and Syria and these fields need to be safeguarded in order to prevent acts of vandalism which are causing the complete loss of local varieties [37,93]. Moreover, countries from Central Asia where pistachio originated, possess pistachio germplasm with high lipid content, wide tolerance to abiotic stress (i.e., heat, drought, and cold) which can be used in future pistachio breeding programs [2]. Different strategies to conserve these unique germplasms are summarized in the next section.

### 5.3. Pistachio Rootstocks

Except for *P. vera* which with its edible and economically important nuts, the other *Pistacia* spp. have been used as rootstocks or for agroforestry purposes such as *P. chinensis*, *P. lentiscus*, *P. mexicana*, *P. weinmannifolia* and other local species such as *P. aethiopica* (*P. lenticus* subsp. *emargianta)*, *P. palaestina (P. terebinthus* subsp. *palaestina)* and *P. texana* (*P. mexicana*) to increase biodiversity and reduce erosion [90]. Some cultivars of *P. vera* have been used for years as rootstock for establishment of new orchards mainly in Iran, Syria and Turkey. They are stronger, more homogenous and produce more lateral roots and thicker stems which enable them to grow in drought regions containing soils with high lime content and salinity. For instance, in Iran the use of tolerant *P. vera* cvs. ‘Ghazvini’ and ‘Badami-Zarand’ rootstocks (to soils salinity) or ‘Badami-Zarand’ and ‘Sarakhs’ rootstocks (to water deficient areas) are usually recommended [46,103]. However, they are sensitive to high soil moisture as well as soil-borne pathogens such as *Meloidogyne* spp. nematodes and *Phytophthora* spp. pseudofungi [104]. Moreover, incompatibility as a result of growth rate differences between rootstock and grafted-scion occurs frequently when *P. vera* trees are used as rootstock [105,106].

A range of factors need to be considered when choosing a rootstock. These include growth characteristics, incompatibility, cold tolerance, soil-borne pathogen tolerance and salinity followed by horticultural effects, micronutrient uptake and effects on scion yield, quality and alternate bearing. For that, among the wild *Pistacia* species, only six have been widely used as rootstocks for pistachio: *P. atlantica* sub. mutica and *P. khinjuk*, in Iran [61], *P. atlantica*, *P. chinensis* subsp. *integerrima* (generally known as *P. integerrima* Steward ex Brandis), and *P. terebinthus*, in the USA [107] and *P. atlantica*, *P. khinjuk*, *P. terebinthus* subsp. *palaestina* (generally known as *P. palaestina* Boiss.), and *P. terebinthus* in Turkey [108,109], and *P. atlantica* and *P. terebinthus* subsp. *palaestina* in Syria [3]. Recently, interspecific hybrids such as ‘UCB1’, a closed cross ♀ *P. atlantica* × ♂ *P. chinesis* subsp. *integerrima* [110], has been developed. The interspecific hybrid ‘UCB1’ shows exceptional vigor and produces much higher yields than ‘Kerman’, as it anticipates the entry into production of the graft-section much earlier; in fact, production started at 4–5 years, as has occurred in the San Joaquin Valley of California [90], instead of at 8–10 years when other rootstocks are used.

Important physiological properties of some pistachio rootstocks from different aspects have been taken into account in Table 2. Whilst almost all of the rootstocks are vigorously grown, differences can be highlighted among them, related to seed germination abilities, with highest rates in *P. chinesis* subsp. *integerrima* and the lowest in *P. terebinthus* and *P. atlantica* sub. *mutica*; high absorption of mineral nutrients from soil as *P. terebintus* and lowest in *P. chinesis* subsp. *integerrima*; high compatibility with scions such as *P. kinjuk*; high tolerance to cold stresses such as *P. terebinthus* and *P. khinjuk* and finally high resistance to biotic pressures *P. chinesis* subsp. *integerrima*, the interspesific hybrid ‘UCB1’ as well as *P. atlantica* sub. *mutica* [98,108,111,112].

## 6. Pistachio Germplasm Propagation and Conservation

Historically, *Pistacia* spp. germplasms were mainly conserved via conventional macropropagation techniques using in situ (in-site in their native place) habitats or even old orchards, and ex situ (off-site) where material is taken away from their native place to germplasm, botanic gardens, and so on. The strategy is to identify superior genotypes and transfer them to collections as well as to maintain them in the wild [116].

In recent years, unconventional biological techniques, including cryopreservation (for the long-term), slow-growth storage conditions and synthetic seeds (for medium-term) and micropropagation (for short-term) have opened new insights for preservation of commercial and endangered *Pistacia* species [19,20,21,22,23]. It should be noted that although pistachio species are not globally endangered, at least 12 species are currently included in the IUCN Red List of Threatened Species [117]: *P. cucphuongensis* (VU, vulnerable), *P. vera*, *P. aethiopica*, *P. mexicana*, *P. atlantica* (NT, near threatened) and the rest are LN (least concern), so it has been imperative to apply preservation policies to these species [118].

### 6.1. Conventional Sexual Propagation

Pistachio reproduces naturally by cross-pollination, so the seeds produced will be as genetically variable as wild genetic populations. It is believed that the first plantings of pistachios in Iran, Turkey, Syria and most of other countries in their native region, were started from selecting and germinating the best seeds of wild *Pistacia* spp. [37]. However, the use of the seed method for pistachio propagations is not really suitable for commercially establishment of pistachio orchards, because nuts of wild *Pistacia* spp. are so sensitive to desiccation, as shown by the low germination rates [119]. Additionally, the resulting trees from seedlings are expected to be genetically variable, although they appear to be quite uniform in the field [90]. Most of the old commercial pistachio orchards currently in use in Iran, Turkey and Syria are the result of two forms of genetic selection: (i) selecting wild pistachios of *Pistacia* sp. and their hybrids capable of bearing fruit in only 5–7 years (normally taking 20–25 years), for hundreds of years, and (ii) using conventional asexual propagation systems, such as grafting selected pistachio tree scions onto rootstocks which are also selected for their agronomic characteristics. This has led to the establishment of orchards in a much shorter time [2,101].

### 6.2. Conventional Asexual Propagation: Cuttings and Grafting

Domestication of trees using sexual reproduction, by seeds, present some bottlenecks which naturally delay the process because of the need for two compatible parental trees, in the case of dioecious plants as pistachio, or good conditions for wind-pollination, a long period of time before obtaining the next generation of mature adult tree capable of bearing fruit and, finally, propagation through seed germination is not suitable for establishment of clonally elite orchards since seedlings are not true-to-type trees. Alternatively, asexual or vegetative propagation has been used to improve tolerance/resistance to biotic and abiotic stresses and to accelerate pistachio domestication [3]. Among others, cutting [120] and grafting [121] methods for macropropagation of pistachios have been traditionally used [37,97].

#### 6.2.1. Propagation by Cuttings

Cuttings are applied to all those woody species in which rooting is not very difficult. In this sense, a range of factors have been reported to impact on the rooting percentage of cuttings from woody species in nurseries: genotype, position on the shoot from which the cutting is made, auxin type and concentration, date of shoot collection and so on, which have been reviewed elsewhere in detail [122,123]. In *Pistacia* spp. cuttings were traditionally used for the clonal propagation of selected rootstocks in order to obtain uniform progeny [124] (and rarely on cultivars [125]). However, attempts to propagate pistachio rootstocks commercially by soft and hard wood cuttings especially using adult materials have given inconsistent results [125,126].

Pioneering studies on pistachio propagation has shown that the use of pistachio hardwood cuttings, obtained from adult trees, have exhibited a very low rooting rate (about 5%), which has been attributed to the fact that adult trees lose their rooting capacity with age [127,128]. In fact, the first experiments carried out with *P. vera* and two clonal rootstocks *P. palestina* and *P. atlantica* rooted with IBA at 10 mg L^−1^ was completely unsuccessful [129]. In the 1980s, softwood, instead of hardwood, cuttings of young (one-year-old) *P. chinensis* trees supplied with the rooting phytohormone Indole butyric Acid (IBA) at 5 mg L^−1^ resulted in 92% rooting [130]. In addition, high rooting percentages (78–100%) were obtained from *P. vera* after dipping the softwood cuttings from seedlings into concentrated 500 and 1000 mg L^−1^ IBA, respectively. Moreover, promising high rooting percentages (88%) of softwood cuttings were obtained using a mist system combined with treatments with IBA at 35 mg L^−1^ [125]. However, only a 50% rooting was achieved if older plant material (four-year-old) was used [131]. Finally, Almehdi and co-workers documented the significant effects of genotype, part of the shoot from which the cutting was taken and shoot collection data on successful rooting percentage (40%) of the rootstock ‘UCB1’ cuttings, but no effect of IBA concentration and shoot lengths was found [132].

#### 6.2.2. Propagation by Grafting

Grafting has been used in pistachio domestication as far as 3000–4000 years ago [133,134], being the decisive horticultural technique in the spread of pistachio cultivation from Central Asia to the countries of the Mediterranean basin between 1000 B.C. and 1000 A.D. [6,135]. From a biological point of view, grafting can be defined as the fusion into a single organism of a portion of tissue from one plant with a portion of tissue from another plant, which can occur naturally and artificially [136]. From a horticultural point of view, it is a technique of cloning by vegetative (asexual) artificial propagation of plants in which a portion of tissue (called scion) from a selected cultivar (donor) is attached to another (called stock or rootstock) selected plant for its resistance to biotic and abiotic stresses and serves as the root system of the grafted plant, so that both grow as a single and new organism. Finally, from a genetic point of view, grafting implies the formation of a new organism through the fusion of at least two genotypes, each one maintaining its own genetic identity in the new grafted plant [135].

Grafting has been used in the last 3500 years for various biological and horticultural purposes such as vegetative macropropagation, juvenility avoidance, cultivar change, to repair established adult trees, size-controlling grafted plants, physiological studies, biotic and abiotic stress resistances or identification of asymptomatic viruses [121,137,138,139]. In the case of pistachio, it was traditionally used for vegetative macropropagation and genetic improvement using *P. vera* plants as a scion and wild *Pistacia* spp. as rootstocks, as described above.

#### 6.2.3. Propagation by Budding

Budding is a special type of grafting, in which a small piece of shoot with a single vegetative bud is cut from the wood of the scion and transferred to the rootstock. To improve the efficiency of budding in pistachio species, a number of factors must be taken into account: the selection of a healthy, active rootstock in optimum condition, with a diameter large enough to adequately accommodate the grafted buds, the quality of the scion wood, the use of the most practical technique for budding (e.g., T-budding in the case of pistachio), compatibility between scion and rootstock, the budding season, and proper aftercare [120,140,141,142].

In general, although several improvements based on those traditional techniques of conventional asexual propagation have been reported, low rooting efficiency, lack of producing uniform plants with a high quality at sufficiently large scales were the most important challenges of pistachio nurseries [141,143], and new strategies for conservation using novel unconventional propagation methods based in plant tissue culture were developed for pistachio.

### 6.3. Unconventional Asexual Propagation

#### 6.3.1. Cryopreservation

Cryopreservation can be classified as a long-term conservation technique based on the methods of cooling and storing biological material at very low temperatures in liquid nitrogen for an unlimited time. Specifically, plant cryopreservation refers to the storage of any plant biological material (cells, tissues, or organs) such as seeds, pollen, shoot tips or dormant buds at an ultra-low temperature in order to decrease by a state of non-division and zero metabolism without any genetic alteration or modification in the plant material for an unlimited period of time [144,145,146]. Recently, Sharrock stated the cryopreservation technique as a basic conservation tool for germplasm conservation in the last Global Strategy for Plant Conservation Report [147]. General procedures for the cryopreservation technique include [118,144]: (1) cold hardening (4 °C generally), (2) pre-culture with suitable cryoprotectant (such as sucrose), (3) physical dehydration, over silica gel, or chemical dehydration by using highly concentrated plant vitrification solutions (i.e., PVS2), (4) freezing by immersion in liquid nitrogen (−196 °C), (5) rapid thawing at 35–40 °C and (6) washing and re-culture into fresh medium.

In the case of cryopreservation of *Pistacia* germplasm, the former technique was developed successfully for seeds of *P. vera*, *P. lentiscus* and *P. terebinthus* [148] and embryonic axes of *P. vera* [149]; whilst the latter technique was utilized for cryopreservation of the in vitro cultured axillary buds of *P. vera* [20] and [150]. Despite this, problems related to the low efficiency of recovery of cryopreserved plant materials makes the application of long-term conservation for *Pistacia* germplasm conservation costly [19]. Thus, much research remains to be conducted in order to design effective cryopreservation protocols that can be used for a wide range of *Pistacia* species in a more efficient and less costly manner.

#### 6.3.2. Slow-Growth Storage

Slow-growth storage can be classified as a medium-term conservation technique and consists of maintaining in vitro plant cultures under limited growth conditions, allowing subculture intervals to be extended up to a few months or a few years, without affecting their potential for survival and regeneration after storage [151]. Slow growth protocols require modification of the usual in vitro culture conditions by various means, such as reducing the incubation temperature (but always keeping it above 0 °C) [152], light intensity or even eliminating it completely [153], nutrient concentration in the culture medium and/or supplementation of the culture medium with different osmotic regulators, commonly sugars (sucrose) or alcohols (mannitol or sorbitol) [154] and/or growth retardants (abscisic acid) to minimize the growth of in vitro conserved plants [155,156].

Barghchi, in 1986, pioneered the development of an efficient protocol to increase subcultures interval in pistachio to 18 months by reducing the growth chamber temperature from 26 °C to 4 °C, applying readjustments in photoperiods and light intensity, supplementing the basal culture media with growth retardants such as the phytohormone abscisic acid or the osmotic mannitol at different concentrations [157].

More recently, Kocand co-workers induced cold storage by reducing the temperature of incubation to 4 °C in dark for 2–12 months. They found that during the first 6 months the pistachio plantlets remained green and healthy but after that the occurrence of shoot tip necrosis and death of the shoots in the cold stored plants was found; after 6 months the cold stored plantlets showed more genetic instability than the controls [158]. The studies conducted on slow-growth storage for many plant species are still very limited [159], including *Pistacia* spp. [20], due to almost all species requiring an appropriate conservation strategy.

#### 6.3.3. Synthetic Seed Techniques

Another unconventional asexual propagation technique considered as medium-term conservation technique is called synthetic seeds or encapsulation, which is a biotechnological tool based on plant tissue culture that has been widely used to propagate, conserve and deliver germplasm of many economically important species [160,161,162]. It consists of artificially constructing synthetic seeds by encapsulating any somatic plant cells, tissue or organs (i.e., somatic embryos, apical or axillary buds, nodal segments, etc.) in gel capsules, forming spherical beads, for conservation [163]. The synthetic seed technology has led to encapsulated, storage and re-growth clonal plant material (alive somatic explants) allowing the possibility of mass production of elite plant species, and even the automatization of this process. Moreover, synthetic seeds show several advantages with respect to conventional propagation systems compared to seeds, cuttings or grafting, because the encapsulated explants upon germination behave like seeds and possess the ability to convert into plantlets under either in vitro or in vivo conditions [164,165]. Although synthetic seeds are a very promising technology, the protocol for the processing of synthetic seeds present several limitations: limited control of somatic embryogenic genesis, developing and maturation and, once encapsulated, the control of seed germination and production of viable explants to be used as plant material. All of these factors need further optimization [166].

In pistachio, the first synthetic seeds were developed using somatic embryos and embryogenic masses in 1996 by Onay and colleagues [167]. Later, pistachio axillary buds were excised from juvenile seeds [150] and shoot tips apices were also encapsulated [20]. Akdemir and co-workers [20] reported that encapsulated shoot apices of *P. vera* cultivars were conserved for up to 12 months at 4 °C in dark conditions with the subsequent recovery of over 90%.

#### 6.3.4. Micropropagation

The conservation of *Pistacia* spp. germplasm by conventional macropropagation methods has some limitations, such as low rooting efficiency, lack of production of high-quality uniform plants at sufficiently large scales, poor seed viability (after conventional storage), high risk of disease spread, and loss of genetic germplasm, as mentioned above and elsewhere [152,168]. To overcome those limitations, in the last 35 years, several unconventional asexual plant propagation methods based on plant biotechnological advances have been applied for pistachio propagation, including the development of in vitro propagation techniques such as organogenesis [169], somatic embryogenesis [170] and micrografting [131,171,172,173]. These have been widely used for mass propagation and germplasm conservation of *Pistacia* species. There are many good reviews about the principles of in vitro culture techniques; here, only very general definitions and principles, and physiological problems related to micropropagation are summarized.

Micropropagation is an in vitro plant tissue culture technique used to multiply a large number of identical selected plants (clones) from a plant stock in a very short time, at a competitive price and with high survival rates. Micropropagation is also known as true-to-type in vitro propagation. It has been used for multiple purposes, such as providing quality plants in large quantities (mass propagation) to meet horticulture, silviculture, genetic conservation and industrial (food, pharmaceutical, nutraceutical and cosmeceutical) needs [174]. Micropropagation has many advantages over conventional propagation systems; among others, the possibility of a massive multiplication of elite genotypes in a very short time, the rescue of those genotypes in danger of extinction and those others difficult to propagate by conventional techniques; all these advantages are easily applicable to the *Pistacia* genus, as described below [175]. The success or failure of micropropagation methods, regardless of whether they are initiated from apical or axillary meristems, adventitious shoots or somatic embryos, depends critically on the effect of a large number of factors, including genotype, culture conditions and, above all, the composition of the culture medium (mineral nutrients of the culture medium, PGRs and vitamins) [176,177,178,179].

In *Pistacia* spp. most of the micropropagation studies have used Murashige and Skoog (MS) [180] as a basal culture medium for the cultivar *P. vera*; [23,181,182,183,184,185] and for several rootstocks such as *P. khinjuk* [119,186,187] *P. atlantica* [131,186], *P. lentiscus* [188], and, less extensively, other basal culture media designed for woody plants such as DKW [189] for *P. vera* [184] or WPM [190] for *P. vera* [169,171,191,192,193]. However, micropropagation of *Pistacia* sp. has been hampered by a low multiplication rate in addition to some physiological disorders such as basal callus (BC), shoot tip necrosis (STN), hyperhydricity (H), Leaf yellowing (LY), or vascular necrosis (VN), when these three basal culture media have been used [106,169,192,194]. In addition, some common problems to micropropagation such as culture browning (CB), caused by phenolic compounds secretions, and contamination (CO) were detected in pistachio micropropagation. Table 3 summarizes the main in vitro studies carried out on *P. vera* L. cultivars and on the different rootstocks, describing the main problems and common physiological disorders observed and the solutions applied to alleviate them; however, those readers interested in knowing the specific solution to each problem and disorder can consult them in detail in each of the referenced studies.

Recent studies on pistachio in vitro propagation growth response to mineral nutrients of culture media have revealed the key roles of some ion interactions (SO_4_^2−^ × Cl^−^, K^+^ × SO_4_^2−^ × EDTA^−^, and Fe^2+^ × Cu^2+^ × NO_3_^−^) for shoot quality, proliferation rate, and shoot length, whilst physiological disorders (BC and STN) were found to have been significantly impacted by independent ions as Fe^2+^ and EDTA^−^, respectively [201]. To alleviate STN of *P. vera* L., some authors have recommended an increase in both the MS boron concentration (up to 100–1000 µM) and calcium concentration (15 mM calcium gluconate [192] or 12–24 mM CaCl_2_·2H_2_O [194]). On the other hand, depletion of MS NH_4_NO_3_ and CaCl_2_ (10.33 mM and 1 mM, respectively) and enhancing FeSO_4_·7H_2_O and Na_2_EDTA·2H_2_O (both 0.15 mM) has been recommended to control STN and LY (chlorosis) in *P. vera* cv. ‘mateur’ [193].

Early studies on the effect of up to 48 µM AgNO_3_ have demonstrated improved shoot regeneration and growth and strong anti-browning effect on culture media and explants in some *Pistacia* species [197,203]. However later on, media supplemented with AgNO_3_ at 24 µM caused some deteriorative effects on the proliferation rate of *P. vera* rootstock, although BC was prevented. The most recent studies have revealed the negative effect on some growth parameters of NO_3_^−^, Mg^2+^, Ag^+^ and gluconate^−^ at high concentrations [204].

To improve pistachio shoot multiplication, standard culture media vitamins have been replaced with other vitamins [186] or a mixture of them [193,197]. In some pistachio studies [185,186,198], MS-vitamin has been replaced by Gamborg B5-vitamin mixture [205]. Some amino acids, such as L-valine and L-Cystein have been employed by some authors as alternative vitamins during pistachio micropropagation [188,197]. Interestingly, among standard MS-, DKW- or B5-vitamin mixtures, nicotinic-acid and pyridoxine-HCl were characterized as having a positive and significant effect, improving shoot length and the total healthy fresh weight of pistachio rootstock [204]. More recently, different physiological disorders in *Pistacia* were subjected to study by our group using computer-based tools, revealing the requirement for high concentrations of thiamine-HCl to alleviate leaf necrosis and leaf color disorders [202].

Recently, the application of RITA^®®^ as an alternative approach to semi-solid media has enabled researchers to develop efficient protocols for mass-propagation of economically important plant species. In recent years, micropropagation of pistachio buds was improved using RITA^®®^ through immersion of nodal and apical explants for 24 min every 16 h in MS medium containing 4 mg L^−1^ BA and 0.1 mg L^−1^ GA_3_ while reducing hyperhydricity and STN symptoms [186]. Therefore, RITA^®®^ could be considered for the mass propagation of pistachio and its rootstocks.

## 7. Computer-Based Tools for Efficient Micropropagation and Conservation of *Pistacia* sp. Germplasm

Numerous protocols for plant tissue culture have been set up since the introduction of the concept of plant cell totipotency by Gottlieb Haberlandt [206], a hypothesis of global dimensions that represents the true starting point of tissue culture [207]. Despite this, the formulation of efficient culture media for any plant species has been the main challenges for plant biotechnology researchers worldwide. For instance, many authors articulated protocols of micropropagation of different *Pistacia* sp. through different approaches such as readjusting the composition of the mineral nutrients of standard culture media, plant growth regulators, applying different culture systems, container type, temperature, light intensity/quality, etc. [208], which varies incredibly from one species to another. The following diagram of Ishikawa represents the most commonly used factors of the in vitro multiplication process (Figure 2):

In order to understand the effect of multiple factors on complex systems, some computer-based tools such as DoE and machine learning algorithms have been recently used in micropropagation studies [176,201].

### 7.1. Design of Experiments (DoE) Tools

Niedz and Evens [209] described DoE as a knowledge-based rather than data-driven technique by integrating true measures of quality and productivity within the experimental process itself. They also pinpointed the great advantage of DoE to other traditional design approaches by simultaneously minimizing data quantity and maximizing data quality on the basis of ‘hierarchical ordering’ and ‘sparsity of effects’. In other words, the DoE approach focuses on identifying the minimum number of factors that cause a significant effect on a process, whereas the general thinking is that it is necessary to determine all factors, and therefore all effects, to fully understand any multifactorial process. Thus, DoE is based on the assumption that only low orders (hierarchical ordering) with a few factors (sparsity of effects) are necessary and sufficient to effectively understand complex processes. Advantages of using DoE versus other traditional strategies such as ‘trial and error’, ‘one factor at time (OFAT)’, or ‘triangulation method’ to improve knowledge about the effect of media components on micropropagation of healthy plants, have been described in that study and elsewhere [176].

Following this hypothesis, the DoE approach was recently used to achieve a deeper understanding of the effect of macro- and micro-nutrients of MS on optimal growth of two pistachio rootstocks, *P. vera* cv. ‘Ghazvini’ and ‘UCB1’, by reducing the experimental design space from 3125 to just 29 treatments [201]. In particular, DoE was utilized to create a five-dimensional design space by categorizing MS salts into five independent factors (NH_4_NO_3_, KNO_3_, mesos (CaCl_2_, MgSO_4_ and KH_2_PO_4_), micro-elements and iron at five concentrations. Later, DoE was utilized to reduce the number of treatments from 6250 to only 61, generated in a five-dimensional IV-design space. In addition, the application of neurofuzzy logic revealed cause-effect relationships between the factors studied (25) and some physiological disorders [202].

### 7.2. Artificial Neural Networks Tools

The field of artificial intelligence (AI) was founded at a conference held at the campus of Dartmouth College, Hanover University during the summer of 1956. Since then, a broad range of studies have been carried out to understand the learning capacity of humans to create a compatible non-biological systems that can achieve its capacity [210]. The main objective of AI was the development of machines with the capacity of learn from data, by using symbolic methods, mainly neural networks. The basal architecture of any artificial neural networks are the individual processing elements called perceptron (mimics of biological neuron) [210]. In the 1990s, a subfield of AI named Machine Learning (ML) emerged, based on an ability to learn. ML algorithms were designed to develop models to first classify data (training) and later to make predictions (testing). Those ML algorithms used methods derived from statistics, fuzzy logic and probability theory [211]. The main difference between AI and ML is that AI needs an active external agent to learn and perform concrete actions that increase the chances of achieving the proposed objective, while ML is able to learn and predict based on passive observations [212].

Among different tools of AI, Artificial Neural Networks (ANNs) and ML, Fuzzy Logic (FL),algorithms are considered the most important scientific advances of recent years with a range of important applications in understanding plant complex bioprocesses [210,213,214], showing many advantages over traditional statistics [215,216]. Different combination of ANNs plus FL and Genetic Algorithms have been used to model complex in vitro processes such as shoot multiplication [217], rhizogenesis and acclimatization [218], seed germination [219], designing plant culture media [178,220] and elicitation of bioactive compounds accumulation [221].

ANNs in combination with fuzzy logic (named as neurofuzzy logic) or with genetic algorithms (ANNs–GA) have been used as modeling tools to achieve an insight, predict and optimize the effect of several independent factors on four growth parameters during micropropagation of *P. vera* [204]. Briefly, twenty-six media ingredients, including mineral nutrients, vitamins and PGRs in different concentrations, were included as inputs and some growth parameters such as proliferation rate, shoot length, etc. as outputs in the neurofuzzy logic model. The IF-THEN rules from neurofuzzy logic models uncovered the positive (BAP, nicotinic-acid and pyridoxine-HCl) and negative (NO_3_^−^, Mg^2+^, Ag^+^ and gluconate^−^) effects on the growth parameters and the critical role of BAP overall growth and physiological parameters. In addition, a pistachio-specific in vitro culture medium, named Pistachio Optimized Medium (POM), was designed by combining ANNs and GA technology. The POM media predicted better growth performance and less physiological disorders than any other medium previously used for pistachio micropropagation. These predictions were experimentally verified and validated, demonstrating that POM simultaneously maximizes growth parameters and minimizes physiological disorders [204].

The combination of ANNs and DoE have also been used successfully to decipher the key role of some ion interactions (SO_4_^2−^ × Cl^−^, K^+^ × SO_4_^2−^ × EDTA^−^, and Fe^2+^ × Cu^2+^ × NO_3_^−^) for shoot quality, proliferation rate, and shoots length [201] and to reveal physiological responses of *Pistacia* to in vitro culture media ingredients [202].

In conclusion, the application of new computational tools, such as DoE and ANNs, is very useful in the development of specific culture media for pistachio efficient micropropagation and conservation, including threatened *Pistacia* germplasm which could be used as cultivars and rootstocks. Such tools help to reduce treatment combinations by minimizing those required, while maximizing data quality (DoE) and deciphering the key factors affecting each parameter in line with the development of optimized culture media (ANNs).

## 8. Conclusions and Future Perspectives

The pistachio is a well-known and widely consumed fruit around the world, but many people are still unaware that all the names by which the pistachio is called in different languages come from a single word “pstk”, pronounced as “pistag” in Avestan, an ancient language of the Persian Empire. Moreover, many states claim to be the area of origin of the pistachio, and several archaeological studies describe it as originating in the arid areas of ancient Mesopotamia, others in present-day Iran, Turkey, Syria and other countries of the Mediterranean basin, but more recent studies show its origin in southern Central Asia, including northern Afghanistan and northeastern Iran. Another peculiarity of the pistachio is that it has been consumed for some 300,000 years, with evidence of its consumption by Neanderthals. Another, no less interesting aspect, is the excellent organoleptic and health properties of the pistachio nuts, as it is very rich in both essential food components (unsaturated fatty acids, carbohydrates, proteins, dietary fiber, vitamins and minerals,) and bioactive phenolic compounds which promote human health due to their antioxidant capacity and oxidative stability, anti-inflammatory, antifungal, antimicrobial and antiviral properties, anti-anxious and depressive behaviors, modulating the intestinal microbiota.

Although most people believe that Iran, where pistachios are called “green gold”, is the world’s largest producer and exporter, the truth is that it is actually the United States, specifically the state of California, followed by Turkey, Iran, China and Syria, which account for 99% of world production; but USA and Iran export the 82% of the pistachio consumed worldwide. However, a large proportion of the pistachio orchards in those countries have been established based on a monoculture systems using a single highly productive species: *P. vera* and two main cultivars: Peter (male) and Kerman (female), while annually a high biodiversity of cultivars and wild *Pistacia* sp. are highly threatened. Numerous pistachio germplasms, cultivated for local uses such as fruit consumption, oil extraction and soap production, are under continuous and extreme genetic erosion, and are threatened by biotic and abiotic damage and anthropogenic pressure; today, they are seriously threatened. In other words, many of the ancient cultivars (700–1800 years old) and wild species of *Pistacia* are in serious danger of extinction, so it is urgent to search for new highly efficient conservation and propagation systems, based on the combined application of biotechnological propagation techniques (in vitro culture) and computational tools. In this sense, our group has been a pioneer in the micropropagation of both *P. vera* and the UCB1 rootstock, designing efficient micropropagation methods and combining DoE and machine learning tools (artificial neural network algorithms) that could be used for ancestral species and hybrids of cultivars and rootstocks of *Pistacia* and for other economic, social and therapeutic species of woody plants.

## Figures and Tables

**Figure 1 plants-12-00323-f001:**
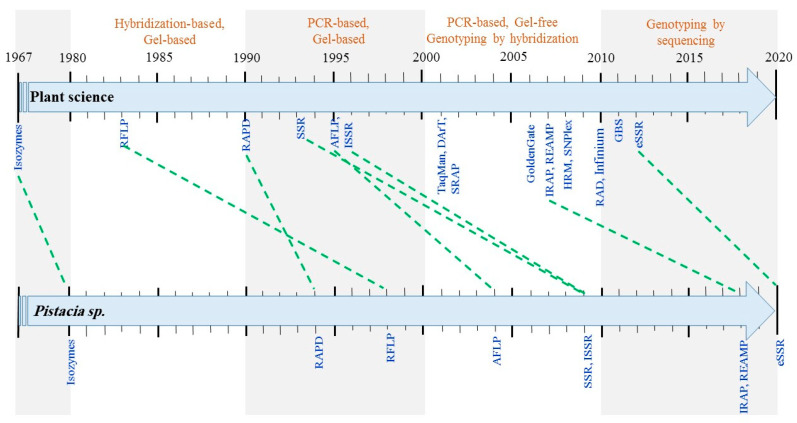
Timeline for the first scientific report for the employment of different commonly used markers in plant science and genus *Pistacia* from isoenzymes to modern molecular makers such as RFLP, RAPD, AFLP, SSR, ISSR, IRAP, REAP and GBS (for detail description see text below).

**Figure 2 plants-12-00323-f002:**
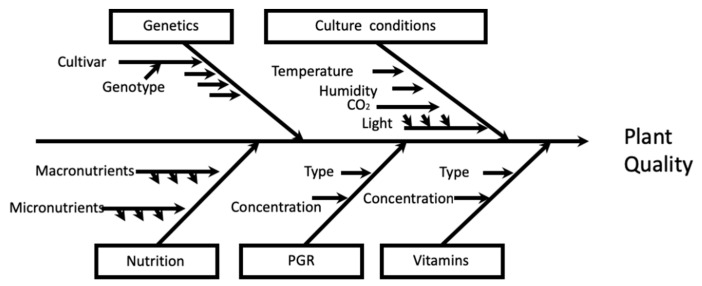
Ishikawa diagram representing the main group factors related with the plant material, in vitro culture conditions, and media composition (minerals, plant growth regulator and vitamins) affecting micropropagated plant quality.

**Table 1 plants-12-00323-t001:** Morphological identification and division of main *Pistacia* species based on leaves characteristics in three sections (modified from Al-Saghir and co-workers [24]).

*Pistacia* sp.	Type	Size	Petiole	Rachis	Leaflets
*Pistacia Section*					
*P. atlantica*	D, M and I	L	F-	NW	(5-)-7–11, L and O
*P. chinensis* subsp. *chinensis*	D, M and P	L	A or F	None	8–14, becoming G and L
*P. chinensis* subsp. *falcate*	D, M and I	L	F	None	11–15, G and L
*P. chinensis* subsp. *integerrima*	D, M and P	L	A or F	None	6-(9)-10, G and L
*P. eurycarpa*	D, M and I	L	F	NW	(1–3-)-5–7, margin G and L
*P. khinjuk*	D, M and I	L	A or R	None	1–9, A and G
*P. terebinthus*	C, D, I and P	L	R	None	(3-)6–11, G and M
*P. vera*	C, D, and I	L	F	None	3–5, M or O
*Lentiscella Section*					
*P. lentiscus* subsp. *lentiscus*	sub C and P	S	F	NW	4–10-(-12), G and M
*P. lentiscus* subsp. *emarginata*	C and P	S	F	W	6–16, E and G
*P. Mexicana*	M and I	S	F	NW	10–30, G and M
*P. weinmannifolia*	M and P	S	F	NW	(4–6)12–20, M
*Inter-specific hybrid*					
*P.* × *saportae*	C and I	L	-	NW	7–9, G and M

Leaves type: C (coriaceous); D (deciduous); M (membranaceous); I (imparipinnate); P (paripinnate). Leaves size: Large (8–23 cm long, 5.2–23 cm wide) and Small (2–15.1 cm long, 1.8–10 cm wide). Leaves rachis: W (winged), NW (narrowly Winged) and None (No winged). Leaves petiole: F (flattened); R (rounded) and A (angled). Leaflets: A (acuminate); E (emarginate); G (glabrous); L (lanceolate); M (mucronulate); O (obtuse).

**Table 2 plants-12-00323-t002:** Physiological main characteristics of most important pistachio rootstocks.

*Pistacia* Species	Seed Germination	Growth	Mineral Absorption	Scion Compatibility	Cold Tolerance	Disease Resistance	Nematodes Resistance	Reference
*P. atlantica*	++	+++	++		++	+	+	[46,90,98,101,113]
*P. atlantica* sub. *mutica*	+	+++		+			+++	[46,98]
*P. chinesis* subsp. *integerrima*	+++	+++	+		+	+++		[90,101,113]
*P. khinjuk*		+++		+++	+++		+	[46,98,108,111]
*P. terebinthus* subsp. *palaestina*		++					+	[46,98,114]
*P. terebinthus*	+	+++	+++		+++	+		[90,101,113,115]
UCB1 (♀ *P. atlantica* × ♂ *P. integerrima*)		+++			++	+++		[90,101,113]

+: low; ++: medium; and +++: high.

**Table 3 plants-12-00323-t003:** Main studies on *Pistacia* spp. micropropagation indicating basal culture media employed, physiological disorders such as basal callus (BC), shoot tip necrosis (STN), hyperhydricity (H), Leaf yellowing (LY), and vascular necrosis (VN) and some problems such as culture browning (CB) and contamination (CO), and the solutions proposed to alleviate them.

Species	Type of Culture	Basal Media	Problems and Disorders	Solutions Proposed	Reference
Cultivars					
*P. vera* L.	Shoot tip Nodal buds	MS	BC, CB, STN	1. Use seedling 2. Successive subculture 3. Remove auxins	[195]
*P. vera* L.	Shoot tip Nodal buds	MS	BC, CB, STN, H	1. Remove apical buds 2. Successive subculture 3. Activated charcoal or polyvinylpyrrolidone	[187]
*P. vera* L.	Seedling shoots	WPM	H, VN	1. Use MS free-vitamin 2. Use WPM	[191]
*P. vera* L.	Shoot tips and nodal buds	MS	STN	1. Increasecytokinins 2. Eliminate auxins	[196]
*P. vera* cv. ‘mateur’	Shoot tips and nodal buds	MS	STN	1. Use of liquid culture 2. Use borone 3. Increase calcium	[192]
*P. vera* cv. ‘mateur’	Shoot tips of grafted seedling	MS	CB, STN, LY, H	1.Successive subcultures 2. Decrease subculture period 3. Decrease BAP	[193]
*P. vera* cv. ‘Antep’	Immature kernels	MS	STN, BC, H	1. Change BAP concentrations 2. Increase Ca^2+^ or BO_3_^−^	[169]
*P. vera* L.	In vitro shoots	MS	STN	1. Increase Ca^2+^ (24 mM)	[194]
*P. vera* cv. mateur	Shoot tips and nodal buds	MS	CB	1. Apply L-cystein 2. Activated charcoal or AgNO_3_	[197]
*P. vera* L.	Nodal buds	MS	BC	-	[198]
*P. vera* L.			STN	1. Increase B or Ca	[199]
*P. vera* L.	Nodal buds		STN, CB	1. Use of ascorbic, citric acid 2. Increase subcultures 3. Substitute BAP for Metatobolin and Kinetin	[106]
*P. vera* L.	Shoot tips Nodal buds	MS	STN, H	1. Optimize RITA^®®^ 2. Decrease cytokinin	[186]
Rootstocks:					
UCB1	Shoot tips of seedlings	DKW	STN, LY, CO	1. Use 2% CO_2_ 2. Increase light intensity 3. Eliminate carbon 4. Increase level of BO_3_ and Zn(NO_3_)_2_	[200]
*P. atlantica*	Shoot tips and nodal buds	MS	CB	1. Apply L-cystein 2. Activated charcoal or AgNO_3_	[197]
*P. atlantica*	Shoot tips and nodal buds	MS	CB	1. Apply L-cystein 2. Activated charcoal or AgNO_3_	[197]
*P. khinjuk*	Shoot tips of seedlings	MS	BC	-	[119]
*P. khinjuk* *P. atlantica*	Shoot tips Nodal buds	MS	STN; H	1. Optimize RITA^®®^ 2. Decrease cytokinins	[186]
UCB1	Shoot tips Nodal buds	MS	BC, STN	1. Readjust iron salts	[201]
UCB1	Shoot tips Nodal buds	MS, POM	BC, STN, LY	1. Readjust mineral nutrients and vitamins	[202]

## Data Availability

Not applicable.

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
