# Peer review of "History, Phylogeny, Biodiversity, and New Computer-Based Tools for Efficient Micropropagation and Conservation of Pistachio (Pistacia spp.) Germplasm"

_plants, 2023, doi:10.3390/plants12020323_

Round 1
Reviewer 1 Report
This review presents the origin, domestication, etymology, taxonomy, phylogeny, germplasm diversity, conservation strategies as well as the application of computational tools and Machine Learning in order an efficient micropropagation protocols for Pistacia genus to be designed.
It is a very interesting review summarizing important information regarding Pistacia species. It is well written and it will be a useful article for designing future experiments for scientists working on Pistacia species.
Author Response
Response: Thank you for your valuable comments and support.
Reviewer 2 Report
The authors, Esmaiel et al., have carefully reviewed the literature relevant to pistachio crop improvement. The review is a significant addition to scientific advancement of molecular progress in pistachio breeding and cultivar development.
I recommend few corrections and suggestions to improve the quality of the manuscript:
Writing needs some attention, too shot paragraphs could be avoided. There are several short paragraphs. They could be merged or expanded to improve understanding.
The abbreviation for not winged and narrow winged looks confused. I suggest to write NW for narrow winged and None (for not winged)
Authors should consider addition of some relevant literature on molecular markers in other significant crops and compare it with progress in pistachio molecular breeding. See below few examples from literature on hylogeny by molecular analysis (RAPD, RFLP, AFLP, SSR, ISSR, IT'S, etc):
Multiplex molecular marker-assisted analysis of significant pathogens of cotton (Gossypium sp.), 2022; Biocatalysis and Agricultural Biotechnology https://doi.org/10.1016/j.bcab.2022.102557 (Cotton); Assessment of genetic diversity and volatile content of commercially grown banana (Musa spp.) cultivars, Hinge et al., Scientific Reports, 2022; https://doi.org/10.1038/s41598-022-11992-1 (Banana); Microsatellite and RAPD analysis of grape (Vitis spp.) accessions and identification of duplicates/misnomers in germplasm collection, Upadhyay et al., 2010 Indian J Hortic Volume 67 Pages 8-15; Microsatellite analysis to differentiate clones of Thompson seedless grapevine, Upadhyay et al., 2010, Ind Journal of Horticulture, Volume 67 Issue 2 Pages 260-263
Figure 1. Please elaborate the caption with significant milestones described in it.
The same applies to figure 2, well described figures help to comprehend manuscript clearly.
Author Response
Comment: The authors, Esmaeil et al., have carefully reviewed the literature relevant to pistachio crop improvement. The review is a significant addition to scientific advancement of molecular progress in pistachio breeding and cultivar development.
Response: Thank you for thorough reading and improving our manuscript. We have addressed all comments point-by-point and corrections have been made according to reviewer’s suggestion in our revised manuscript.
Comment: I recommend few corrections and suggestions to improve the quality of the manuscript:
Writing needs some attention, too shot paragraphs could be avoided. There are several short paragraphs. They could be merged or expanded to improve understanding.
Response: Done. Please see new merged and/or expanded paragraphs (Lines 334-356; 388-405; 444-458; 477-484; 486-499; 512-520; 575-596; 598-607; 613-620; 649-672).
Comment: The abbreviation for not winged and narrow winged looks confused. I suggest to write NW for narrow winged and None (for not winged).
Response: Done. Please see changes in Table 1 and in the caption (Line 229).
Comment: Authors should consider addition of some relevant literature on molecular markers in other significant crops and compare it with progress in pistachio molecular breeding. See below few examples from literature on hylogeny by molecular analysis (RAPD, RFLP, AFLP, SSR, ISSR, IT'S, etc):
Multiplex molecular marker-assisted analysis of significant pathogens of cotton (Gossypium sp.), 2022; Biocatalysis and Agricultural Biotechnology https://doi.org/10.1016/j.bcab.2022.102557 (Cotton); Assessment of genetic diversity and volatile content of commercially grown banana (Musa spp.) cultivars, Hinge et al., Scientific Reports, 2022; https://doi.org/10.1038/s41598-022-11992-1 (Banana); Microsatellite and RAPD analysis of grape (Vitis spp.) accessions and identification of duplicates/misnomers in germplasm collection, Upadhyay et al., 2010 Indian J Hortic Volume 67 Pages 8-15; Microsatellite analysis to differentiate clones of Thompson seedless grapevine, Upadhyay et al., 2010, Ind Journal of Horticulture, Volume 67 Issue 2 Pages 260-263
Response: We highly appreciate this suggestion and those relevant references has been included into the reviewed manuscript (red colored text; Lines: 246-254 and references 57 (Lines 1004-1006); 58 (Lines 1007-1009); 83 (Lines 1070-1072), 84 (1073-1074). Also two new references in other crops have been included: 56 (Lines1001-1003) and 59 (Lines1010-1011)
Also check new paragraph with more specific detailed uses of SSR markers (301-310) on pistachio and new references: 82-89 (Lines 1053-1071)-
Comment: Figure 1. Please elaborate the caption with significant milestones described in it.
Response: Done. Please see new caption (lines 264-265).
Comment: The same applies to figure 2, well described figures help to comprehend manuscript clearly.
Response: Done. Please see new caption (lines 744-746).

Reviewer 3 Report
Dear Authors,
The manuscript entitled “History, phylogeny, biodiversity, and new computer-based tools for efficient micropropagation and conservation of pistachio (Pistacia spp.) germplasm” was revised. It was a very well prepared and very laborious work. It will be a useful resource in the literature, especially for researchers working on Pistacia spp. The manuscript can be published after a few correction.
Kind regards
This corrections described below;
Title
Line 2, ……..biodiversity,and…………
Line 3, ……….micropropagationand……
Line 4, …………chio(Pistacia spp.)…..
Abstract
OK
Keywords
# In general, it should be more correct to avoid the use of the same terms used within the title. It would be appropriate to review the keywords and add the ones that are not included in the title.
# According to the journal, the first letters of the keywords should be lowercase.
1. Introduction
OK
2. Pistachio History
Line 145, “………… Currently, Pistacia sp. are distributed….” should not be italicized,
Line 147, ……. (iv)Sino – Japanian…….
Line 157, ……. Iraq [31].More recent………
Line 169, ……… Mediterranean basin [5]and………
Line 174, ………. climate change[36]………
3. Pistachio taxonomy and phylogeny
Line 181, 187, 202, “… the Anacardiaceae (R.Br.) Lindl. family…” Family names are not italicized,
Line 213, ……… P. palestinaBoiss…..,
Line 250, …… Repeat (SSR)[63,64]…….,
Line 262, ………. sects. Lentiscella and EuLentiscus)…….
4. Pistachio characteristics and production
Line 347, … a (mainly in. According to Food and Agriculture Organization’s report[12])…
# The last access dates of the web links given in the text should be specified.
5. Pistachio biodiversity: cultivars and rootstocks
Line 371, 372, …. pests as Megastigmuspistacea) some fungal canker pathogens of pistachio (Leptosilliapistaciae and Cytosporapistaciae)…. Family names are not italicized,
Line 381, ….. n can be exemplified[16]…..
Line 422, … subsp.palaestina)…………….
6. Pistachio germplasm propagation and conservation
Line 621, …….. for conservation [152].The synthetic…
Line 633, ……… colleagues[156]. Later,….
Line 696, ….. or 12-24mM……..
Line 698, …… (both 0.15mM) was recommended……..
7. Computer-based tools for efficient micropropagation and conservation of Pistacia 724 sp. Germplasm
Line 734, ….. sity/quality, etc.[197], which………
Line 777, ….. neuron) [199].In the 1990s,……
Line 810, …… to in vitro culture media…. should be italicized,
8. Conclusions and future perspectives
OK
Tables
OK
Figures
OK
References
OK
Author Response
Comment: The manuscript entitled “History, phylogeny, biodiversity, and new computer-based tools for efficient micropropagation and conservation of pistachio (Pistacia spp.) germplasm” was revised. It was a very well prepared and very laborious work. It will be a useful resource in the literature, especially for researchers working on Pistacia spp. The manuscript can be published after a few correction.
Kind regards.
Response: Thank you for your detailed review and valuable suggestions to improve this manuscript. All space error were due to combine word (from Windows) and open software in the las version. All changes are highlighted in red color.
Comment: TITLE:
Line 2, ……..biodiversity,and…………
Line 3, ……….micropropagationand……
Line 4, …………chio(Pistacia spp.)…..
Response: Done. Please see red color changes in text (Lines 2-4).
Comment: KEYWORDS: # In general, it should be more correct to avoid the use of the same terms used within the title. It would be appropriate to review the keywords and add the ones that are not included in the title.
# According to the journal, the first letters of the keywords should be lowercase.
Response: The “Keywords” have been deeply reviewed in the revised manuscript according to the reviewer’s suggestion and the journal’s guidelines (See lines 34-36).
Comment: PISTACHIO HISTORY:
Line 145, “………… Currently, Pistacia sp. are distributed….” should not be italicized,
Line 147, ……. (iv)Sino – Japanian…….
Line 157, ……. Iraq [31].More recent………
Line 169, ……… Mediterranean basin [5]and………
Line 174, ………. climate change[36]………
Response: Done. Please see red color changes in text (Lines 145, 147, 157, 169 and 174).
Comment: PISTACHIO TAXONOMY AND PHYLOGENY:
Line 181, 187, 202, “… the Anacardiaceae (R.Br.) Lindl. family…” Family names are not italicized,
Line 213, ……… P. palestinaBoiss…..,
Line 250, …… Repeat (SSR)[63,64]…….,
Line 262, ………. sects. Lentiscella and EuLentiscus)…….
Response: Done. Please see red color changes in text (Lines 181, 187, 199, 202, 213, 257, and 269).
Comment: PISTACHIO CHARACTERISTICS AND PRODUCTION:
Line 347, … a (mainly in. According to Food and Agriculture Organization’s report[12])…
# The last access dates of the web links given in the text should be specified.
Response: Thank you again for your comment. In the revised manuscript the last access has been included (Please, see details in line 360 and also in references line 897). Esmaeil: check the journal rules about how to cite a web reference (for line 901).
Comment: PISTACHIO BIODIVERSITY: CULTIVARS AND ROOTSTOCKS:
Line 371, 372, …. pests as Megastigmuspistacea) some fungal canker pathogens of pistachio (Leptosilliapistaciae and Cytosporapistaciae)…. Family names are not italicized,
Line 381, ….. n can be exemplified[16]…..
Line 422, … subsp.palaestina)…………….
Response: Done. Please see red color changes in text (Lines 383-384, 392 and 432).
Comment: PISTACHIO GERMPLASM PROPAGATION AND CONSERVATION:
Line 621, …….. for conservation [152].The synthetic…
Line 633, ……… colleagues[156]. Later,….
Line 696, ….. or 12-24mM……..
Line 698, …… (both 0.15mM) was recommended……..
Response: Done. Please see red color changes in text (Lines 628, 640, 700, and 702).
Comment: COMPUTER-BASED TOOLS:
Line 734, ….. sity/quality, etc.[197], which………
Line 777, ….. neuron) [199].In the 1990s,……
Line 810, …… to in vitro culture media…. should be italicized,
Response: Done. Please see red color changes in text (Lines 737, 783, and 816).

Round 2
Reviewer 2 Report
The manuscript has been sufficiently improved to warrant publication in the Plants journal.